# Fungal Pathogens of Peach Palm Leaf Spot in Thailand and Their Fungicide Sensitivity

**DOI:** 10.3390/jof11040318

**Published:** 2025-04-17

**Authors:** Prisana Wonglom, Nakarin Suwannarach, Jaturong Kumla, Anurag Sunpapao

**Affiliations:** 1Faculty of Technology and Community Development, Thaksin University, Phatthalung Campus, Pa Payom District, Phatthalung 93110, Thailand; prisana.w@tsu.ac.th; 2Center of Excellence in Microbial Diversity and Sustainable Utilization, Chiang Mai University, Chiang Mai 50200, Thailand; suwan.462@gmail.com (N.S.); jaturong_yai@hotmail.com (J.K.); 3Department of Biology, Faculty of Science, Chiang Mai University, Chiang Mai 50200, Thailand; 4Agricultural Innovation and Management Division (Pest Management), Faculty of Natural Resources, Prince of Songkla University, Hat Yai, Songkhla 90110, Thailand

**Keywords:** emerging disease, leaf spot, peach palm, synthetic fungicides

## Abstract

Peach palm (*Bactris gasipaes* Kunth) is a long-lived tropical palm valued for its edible, nutritious fruits. The cultivation area of peach palm, which was introduced to Thailand for fruit production, has been steadily expanding. Small brown spots that expanded into irregular lesions with dark margins were first observed on *B. gasipaes* seedlings in commercial nurseries in Phetchaburi Province, southern Thailand. To identify the causal pathogens, ten fungal isolates were obtained from symptomatic leaves and subjected to pathogenicity tests, confirming their ability to cause the disease. Morphological and molecular analyses identified five isolates as *Colletotrichum fructicola* (BGC02.2, BGC03) and *C. theobromicola* (BGC01, BGC02.1, BGC04) and five isolates as *Fusarium pernambucanum* (BGF01, BGF02, BGF03, BGF04.1, BGF04.2). Phylogenetic analysis was based on *act*, *cal*, *gapdh*, ITS, and *tub2* regions for *Colletotrichum* spp. and *cal*, *rpb2*, and *tef1-α* for *Fusarium* spp. In vitro fungicide assays revealed that *C. fructicola* and *C. theobromicola* were the most sensitive to carbendazim, mancozeb, and prochloraz, while *F. pernambucanum* was effectively inhibited by mancozeb and prochloraz. This study represents the first report of *C. fructicola*, *C. theobromicola*, and *F. pernambucanum* causing leaf spot disease on *B. gasipaes* in Thailand, providing essential insights for disease management strategies in the region.

## 1. Introduction

Peach palm was introduced to Thailand in 2012 by the private sector; since then, it has been cultivated in several parts of the country. Although peach palm is not a common crop in Thailand, its products do add economic value for Thai farmers. Peach palm (*Bactris gasipaes* Kunth) is a tropical palm species belonging to the Arecaceae family, native to Central and South America. Among the palm trees, the peach palm is considered a domesticated and commercial plant with high economic potential [1]. Peach palm fruits contain a high content of insoluble fiber [2] as well as all essential amino acids [3], with their yellow-red pigmentation providing a rich carotenoid profile [4]. This palm is also cultivated for its heart of palm, which is harvested from the inner core of the tree and used as a vegetable with high carotenoid content [5]. The heart of the peach palm can be harvested without killing the tree, making the cultivation of peach palm a more sustainable option compared to other palms that die after a single harvest. Peach palm is commonly grown in tropical regions with high humidity and rainfall, with annual precipitation ranging from approximately 2000 to 5000 mm [6].

Current climate change is known to affect fungal survivability and infectivity, increasing plant host susceptibility and leading to the emergence of new diseases and outbreaks [7]. Climate change also impacts the rising number of fungal diseases, which affect important crops worldwide [8]. The cultivation of peach palm has been challenged by several diseases, particularly those caused by fungi and fungal-like microorganisms. For instance, bud rot disease caused by *Phytophthora palmivora* has been reported in Ecuador; this affects the palm bud, causing internal rotting and killing the growing points of the palm [9]. In addition, black rot disease caused by *Thielaviopsis ethacetica* has been reported in Brazil, where it affects peach palm fruits, causing them to detach easily from the bunch. The fruit tissues show black coloration, and white to black mycelia have been observed [10]. Leaf spots in oil palm (*Elaesis guineensis*), which belongs to the same family as peach palm, are commonly caused by *Curvularia eragrostidis* and *C. oryzae* [11,12]. Although the causal agents of leaf spot diseases are better known in related palm species, those affecting peach palm (*Bactris gasipaes*) have been rarely documented.

The tropical climate in southern Thailand is conducive to pathogen germination and the spread of disease. Plant disease management often relies on biological, cultural, and chemical methods, which provide practical approaches to disease control. The application of synthetic fungicides is considered the fastest and most effective approach to managing plant diseases [13]. However, the sensitivity of fungi to synthetic fungicides can vary, and frequent high-dose applications may lead to the development of resistance, which must be carefully considered. Understanding the sensitivity of peach palm leaf spot pathogens to commonly used fungicides is essential for developing effective management strategies and mitigating the risk of fungicide resistance.

The precise diagnosis of an emerging disease is the first step in effective disease management. In the present study, therefore, the aim was to identify the pathogens that cause leaf spot disease in peach palm seedlings and to perform a screening of their sensitivity to synthetic fungicides. Precise pathogen identification is essential for developing appropriate disease control methods; such understanding may also contribute to sustainable management practices [14], preserving the productivity and economic value of peach palms in the region.

## 2. Materials and Methods

### 2.1. Sample Collection and Fungal Isolation

A total of twenty symptomatic leaves from different plants were collected from one private nursery in Phetchaburi Province, southern Thailand. The leaf samples were kept in plastic bags and brought to the laboratory, where fungal isolation was subsequently conducted. The tissue transplanting method was used to isolate the fungal pathogen. Leaf samples with both diseased and healthy tissues were cut into small pieces (3 × 3 mm) and surface-disinfected with 0.5% sodium hypochlorite (NaClO) for 3 min. The samples were washed in sterilized distilled water (SDW) and then dried on sterilized filter papers. The samples were then placed on water agar (WA) and incubated at 28 ± 2 °C under natural light for 3 days. Hyphal tips recovered from the samples were taken and transferred to potato dextrose agar (PDA) for further identification.

### 2.2. Pathogenicity Test

A pathogenicity test was conducted using the spore suspension of each fungal isolate of 7-day-old PDA cultures [15]. Conidia were harvested, and an adjusted concentration of ×10^6^ conidia/mL was directly applied by spraying healthy peach palm leaves with approximately 20 mL per plant. Spraying with SDW only was used as a negative control. The experiment was conducted in three replicates and repeated twice. Inoculated plants were covered with plastic bags to maintain humidity and then incubated at ambient temperature with natural light for 7 days, during which time the development of leaf spots was observed.

### 2.3. Morphological Study

Fungal isolates were cultured on PDA and incubated at ambient temperature to observe the growth rate. Macroscopic and microscopic features were observed using an S8AP0 stereomicroscope (Leica Microsystems, Wetzlar, Germany) and a DM750 compound microscope (Leica Microsystems, Wetzlar, Germany), respectively. The morphological characteristics of the hyphae, conidiophores, and conidia were measured (*n* = 20).

### 2.4. DNA Extraction, PCR Amplification, and Phylogenetic Analysis

DNA was extracted directly from each one-week-old fungal isolate cultivated on PDA at 25 °C using a DNA Extraction Mini Kit (FAVORGEN, Ping Tung, Taiwan), following the manufacturer’s protocol. Polymerase chain reaction (PCR) was used to amplify the internal transcribed spacer (ITS) region and partial sequences of actin (*act*), beta-tubulin (*tub*), calmodulin (*cal*), glyceraldehyde-3-phosphate dehydrogenase (*gapdh*), RNA polymerase II second largest subunit (*rpb2*), and translation elongation factor 1-alpha (*TEF1-α*) using the primer pairs ITS4/ITS5, ACT512F/ACT738R, T1/T2, CL1C/CL2C or CAL-228F/CAL-2Rd, GDF1/GPDHR2, RPB2-5F2/RPB2-7cR, and EF1/EF2, respectively, following the protocols described in previous studies [16,17,18,19,20]. PCR products were observed by using 1.5% agarose gel electrophoresis and were then purified using the PCR clean-up Gel Extraction NucleoSpin^®^ Gel and PCR Clean-up Kit (Macherey-Nagel, Düren, Germany) following the manufacturer’s protocol. The purified PCR products were directly sequenced. Sequencing reactions were performed, and the sequences were automatically determined using a genetic analyzer at the 1st Base Company (Kembangan, Malaysia).

The sequences were compared to the GenBank database, using the BLASTn program available at NCBI (http://blast.ddbj.nig.ac.jp/top-e.html, accessed on 14 March 2025). Multiple-sequence alignment was performed using MUSCLE [21] with default settings and improved where necessary using BioEdit version 6.0.7 [22]. Phylogenetic analyses of the representative fungal isolates in this study were conducted using maximum likelihood (ML) to determine their phylogenetic positions with RAxML-HPC2 version 8.2.10 [23,24] on the CIPRES web portal, under the GTRCAT model with 25 categories and 1000 bootstrap replications. FigTree version 1.4.0 was used to illustrate the tree topologies.

### 2.5. In Vitro Test of Synthetic Fungicides Against Peach Palm Leaf Spot Pathogens

To test the sensitivity of synthetic fungicides against peach palm leaf spot pathogens, an in vitro test was conducted on PDA plates. Commercial fungicides with the ability to inhibit ascomycete fungi, namely captan, carbendazim, mancozeb, and prochloraz, were prepared as previously described by Thaochan et al. [25]. Fungicides were dissolved or suspended in SDW and adjusted to the recommended concentrations following the manufacturers’ instructions. An agar plug of each fungal isolate was directly placed centrally on the tested plates; isolates cultured on PDA alone served as control. The tested plates were then incubated at 28 ± 2 °C until the growth of each fungal isolate fully covered the control plate. The experiment was conducted with 3 replicates and was repeated twice. The colony diameter of the fungal isolate was measured and converted to percentage inhibition using the following formula:Percentage inhibition (%) = [(Dc − Dt)/Dc] × 100
where Dc is the colony diameter of the pathogen in control, and Dt is the colony diameter of the pathogen in the treatment.

### 2.6. Statistical Analysis

Significant differences among treatments with synthetic fungicides were subjected to a one-way analysis of variance (ANOVA). Statistically significant differences among treatments were analyzed using Duncan’s multiple range test (DMRT) with a ratio value of *p* < 0.05 by the Statistical Package for the Social Sciences (SPSS version 29) program.

## 3. Results

### 3.1. Symptom Recognitions

Leaf spot disease was observed at a private nursery in Phetchaburi Province, southern Thailand, where peach palms are grown for seedling production. The disease first appeared on 3-month-old seedlings as small, circular brown spots that expanded into irregular lesions with dark-stained margins, turning gray with age. Symptoms also developed at leaf tips, forming concentric dark margins and causing tip blight (Figure 1).

### 3.2. Pathogenicity Test

A total of 10 fungal isolates were recovered from twenty symptomatic leaf samples. By using the spore suspension method, it was found that all isolates caused leaf spots on healthy peach palm leaves, similar to those observed in natural infections (Figure 2). Fungal isolates were re-isolated from the inoculated leaves of the peach palm. The morphologies of BGC01, BGC02.1, BGC02.2, BGC03, and BGC04 matched those of *Colletotrichum* spp. and BGF01, BGF02, BGF03, BGF04.1, and BGF04.2 matched those of *Fusarium* spp. Ten fungal isolates were deposited in the Culture Collection of Pest Management, Faculty of Natural Resources, Prince of Songkla University, Thailand.

### 3.3. Morphological Characteristics

Based on morphological characteristics, the ten fungal isolates were classified into two genera: The first group of isolates had a white and cottony (or aerial) colony appearance with an irregular edge, becoming darker with age. Their average growth rate ranged from 0.65 to 1.01 cm per day. Conidiophores were long and hyaline, with openings. Conidia were cylindrical and non-septate, with rounded ends (Figure 3). The conidial size ranged from 13.46 to 16.67 µm in length and 3.55 to 4.44 µm in width (*n* = 20). Appressoria structures were brown to dark brown and irregular in shape. Sexual morphs were not observed in this study. Morphological characteristics identified this fungus as *Colletotrichum* spp. [26].

The second group of isolates had cottony, pale, or white-gray concentric rings, and the average growth rate ranged from 1.05 to 1.29 cm per day. Hyphae were septate and hyaline. Conidiophores were hyaline, and phialides were observed at their tips. Conidia developed at the tips of the conidiophores. Only macroconidia were observed in this study; these were hyaline, 3–4 septate, and curved, with the size of conidia ranging from 18.74 to 31.64 µm in length and 2.89 to 3.25 µm in width (*n* = 20). Chlamydospores were globose to subglobose (Figure 3). Based on the morphology observed in this study, the fungus was identified as *Fusarium* spp. [27].

### 3.4. Molecular Identification

A BLASTn search indicated that the fungal isolates BGC01, BGC02.1, BGC02.2, BGC03, and BGC04 belonged to the genus *Colletotrichum* (Table 1). The ITS, *gapdh*, *cal*, *act*, and *tub* sequences of four *Colletotrichum* isolates in this study were deposited in the GenBank database, as shown in Table 1. The aligned data set contained 2550 bp, including gaps (ITS: 1–810, *gapdh*: 811–1081, *cal*: 1082–1535, *act*: 1536–1819, *tub*: 1820–2550), with 37 taxa. The outgroup consisted of *C. boninense* and *C. brasiliense* from the *C. boninense* species complex. The sequence alignment had 769 distinct alignment patterns, with 15.08% undetermined characters or gaps. The RAxML analysis resulted in a final ML optimization likelihood value of −8820.7028. A phylogenetic tree is represented in Figure 4. The phylogenetic tree successfully assigned all fungal isolates within the *C. gloeosporioides* species complex. Three isolates, BGC01, BGC02.1, and BGC04, were in the same clade of *C. theobromicola*, which included the type species CBS 124945. This clade formed a monophyletic group with 100 bootstrap (BS) support. Therefore, isolates BGC01, BGC02.1, and BGC04 were identified as *C. theobromicola*. It was found that two isolates, BGC02.2 and BGC03, were assigned to the same clade of *C. fructicola*, which included the type species MFU 090228. Thus, these two isolates were identified as *C. fructicola*. Therefore, the multi-gene phylogenetic analysis was based on the ITS, *gapdh*, *cal*, *act*, and *tub* sequences, following the identification techniques used in previous studies [17,19,28,29].

Based on BLASTn results, the fungal isolates BGF01, BGF02, BGC03, BGF04.1, and BGF04.2 belonged to the genus *Fusarium* (Table 1). Therefore, the multi-gene phylogenetic analysis was based on *tef1-α*, *cal*, and *rpb2* sequences, following the identification techniques used in previous studies [18,20,30]. The *tef1-α*, *cal*, and *rpb2* sequences of all *Fusarium* isolates were deposited in the GenBank database and are presented in Table 1. The aligned data set contained 2205 bp, including gaps (*tef1-α*: 1–682, *cal*: 683–1348, *rpb2*: 1348–2205), with 34 taxa. *Fusarium camptoceras* and *F. neosemitectum* from the *F. camptoceras* species complex were used as the outgroup. The sequence alignment had 389 distinct alignment patterns, with 7.69% undetermined characters or gaps. The RAxML analysis resulted in a final ML optimization likelihood value of −6638.6556. The results indicated that all fungal isolates belonged in the Incarnatum clade of the *F. incarnatum*-*equiseti* species complex (Figure 5). It was found that all fungal isolates were successfully assigned within the same clade of *F. pernambucanum*, which consisted of the type species URM 7559. Therefore, this group of isolates was identified as *F. pernambucanum*.

### 3.5. The Effect of Synthetic Fungicides on the Growth of Peach Palm Leaf Spot Pathogens

In this study, we tested the effect of commercial fungicides at recommended doses based on the manufacturer’s instructions. The effects of captan (40 g/20 L), carbendazim (50 mL/20 L), mancozeb (50 g/20 L), and prochloraz (80 g/20 L) were tested on the fungal growth of both *Colletotricum* and *Fusarium* pathogens using an in vitro test. After 7 days of incubation, the control group showed normal growth in both pathogens, whereas the tested plates showed slow growth in the pathogens. Different synthetic fungicides showed different percentage inhibitions, as shown in Figure 6 and Figure 7. *Colletotrichum fructicola* and *C. theobromicola* were inhibited by carbendazim, mancozeb, and prochloraz with percentage inhibitions of 80.00–88.76%, 52.77–90.00%, and 68.00–80.00%, respectively (Figure 6), while *F. pernambucanum* was inhibited by mancozeb and prochloraz with percentage inhibitions of 52.27–89.00% and 78.89–88.00%, respectively (Figure 7). Carbendazim, mancozeb, and prochloraz were the synthetic fungicides that most effectively inhibited the growth of *C. fructicola* and *C. theobromicola*. However, mancozeb and prochloraz were the most effective against *F. pernambucanum*.

## 4. Discussion

This study presents the first global report of *Colletotrichum theobromicola* and *Fusarium pernambucanum* as causal agents of leaf spot disease on peach palm and the first record of this disease in Thailand. We examined these species through both morphological and molecular analyses and tested their pathogenicity on peach palm seedlings to fulfill Koch’s postulates. However, the symptoms observed in the field appeared to be more severe than those shown in pathogenicity tests. This difference might have been due to environmental factors, such as changes in temperature, humidity, and precipitation patterns, which can influence disease severity [31].

The precise diagnosis of fungal pathogens causing plant diseases is the first step in plant disease management. After this, an appropriate control method is selected. In the past, morphological study was the classic method for identifying fungal pathogens [32]. However, due to overlapping microscopic properties, species-level identification can be difficult using this technique. The use of both morphology and modern biotechnology (multiple DNA sequences) is now an effective method for identifying fungi that cause plant diseases. *Colletotrichum theobromicola* was previously identified as the causal agent of anthracnose leaf spot on *Centrosema pubescens* based on morphology and multilocus DNA analysis (*act*, *cal*, *gapdh*, ITS, and *tub2*) [33]. Our findings align with those of Pakdeniti et al. [33], as we successfully identified *C. theobromicola* as the pathogen responsible for peach palm leaf spot using the same approach. Similarly, *Fusarium pernambucanum* was reported as the causal agent of leaf yellow spot in melon based on ITS, *tef1-α*, and *rpb2* sequences [34]. In this study, we identified *F. pernambucanum* as a pathogen of peach palm leaf spot using morphology and the same molecular markers. These results confirm that integrating morphological and molecular analyses is effective for the species-level identification of fungal pathogens.

Recently, plant diseases caused by fungi in the phylum Ascomycota have increasingly been reported in various plant species in southern Thailand [15]. *Colletotrichum theobromicola* has been reported to cause anthracnose on *Anthurium* sp. [35], on *C. pubescens* [33], and on *Cyclamen persicum* [36]. *Colletotrichum fructicola* has been documented as a pathogen causing anthracnose and leaf spot diseases in economically important crops, including tea (*Camellia sinensis*), mango (*Mangifera indica*), and rubber tree (*Hevea brasiliensis*) [37,38,39]. The co-occurrence of *C. fructicola* and *C. theobromicola* on peach palm is consistent with previous reports of *Colletotrichum* species co-infecting the same host [19], suggesting that mixed infections may influence disease severity and progression. Similarly, *Fusarium pernambucanum* has been reported as the causal agent of fruit rot in melon [34] and postharvest fruit rot in mango [40]. However, its presence in Thailand has not been previously documented on any plant host. In contrast, *C. theobromicola* was first reported in Thailand as a pathogen of anthracnose leaf spot on *Centrosema pubescens* [33], but it has not been associated with peach palm disease until now. Globally, there have been no previous reports of *C. theobromicola* or *F. pernambucanum* infecting peach palm. This study provides the first evidence of *C. theobromicola*, *C. fructicola*, and *F. pernambucanum* causing leaf spot disease on peach palm in Thailand and worldwide, highlighting the novelty and significance of these findings. Additionally, the identification of these pathogens expands the known host range of these fungal species and underscores the need for further research on their epidemiology and management. The occurrence of *C. fructicola*, *C. theobromicola*, and *F. pernambucanum* in peach palms raises concerns about their potential spread to other palm species cultivated in Thailand. Moreover, the presence of multiple fungal species in peach palm may indicate complex disease dynamics that require further investigation.

The use of synthetic fungicides with different modes of action has been reported as the fastest and most effective method for controlling plant diseases caused by fungi worldwide [13]. For instance, carbendazim, a systemic fungicide in the benzimidazole group, was previously reported to suppress the fungal growth and spore germination of *N. cubana*, which causes novel leaf fall disease in rubber trees and also enhances the defense response in rubber trees [25]. A reduced incidence of anthracnose-twister was observed in onion following treatment with captan and paclobutrazol against *C. gloeosporioides* and *F. fujikuroi*, while spray applications of carbendazim and paclobutrazol resulted in the lowest degree of disease severity [41]. Furthermore, propiconazole, in combination with prochloraz in the triazoles group, was shown by in vitro test to successfully inhibit *F. proliferatum* and *F. oxysporum*, which cause garlic dry rot [42]. Our findings align with previous studies indicating that carbendazim, mancozeb, and prochloraz—classified as a DNA biosynthesis inhibitor, a multi-site activity fungicide, and a demethylation inhibitor (DMI), respectively—exhibited strong antifungal activity against *C. fructicola* and *C. theobromicola*. Meanwhile, mancozeb and prochloraz effectively inhibited *F. pernambucanum* in vitro, whereas carbendazim showed no inhibitory effect against *F. pernambucanum*, possibly due to the intrinsic resistance in *Fusarium* species [43]. However, the inhibition rates of the tested fungicides against *F. pernambucanum* were relatively low, ranging from 2 to 21%. It is important to note that the fungicide sensitivity assay conducted in this study represents a preliminary in vitro screening. While prochloraz demonstrated the highest efficacy against both pathogens, further validation under field conditions is necessary to determine its practical effectiveness. Additionally, future studies should explore the potential of combining multiple fungicides to enhance disease management strategies for peach palm leaf spot.

## 5. Conclusions

Based on morphological and molecular characterization, this paper is the first report on *C. fructicola*, *C. theobromicola*, and *F. pernambucanum* as causal pathogens associated with leaf spot disease on peach palms in southern Thailand. The synthetic fungicide prochloraz was found to effectively suppress the growth of three fungal species, *C. fructicola*, *C. theobromicola*, and *F. pernambucanum*, in vitro, and this knowledge should enable farmers to quickly bring this disease under control in plant nurseries. The occurrence of these pathogens may have implications for local regional agriculture, emphasizing the importance of integrated disease management approaches. Further research on disease management needs to be conducted to mitigate losses.

## Figures and Tables

**Figure 1 jof-11-00318-f001:**
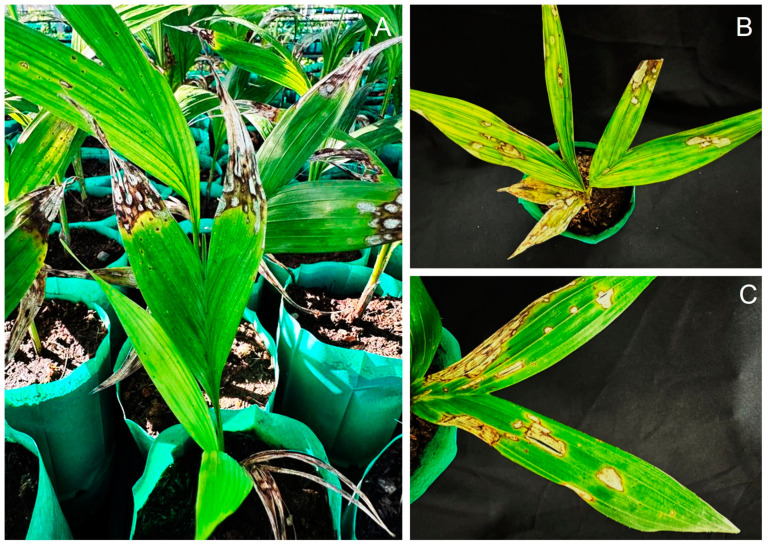
The symptoms of leaf spot on peach palm observed in a private nursery (**A**–**C**).

**Figure 2 jof-11-00318-f002:**
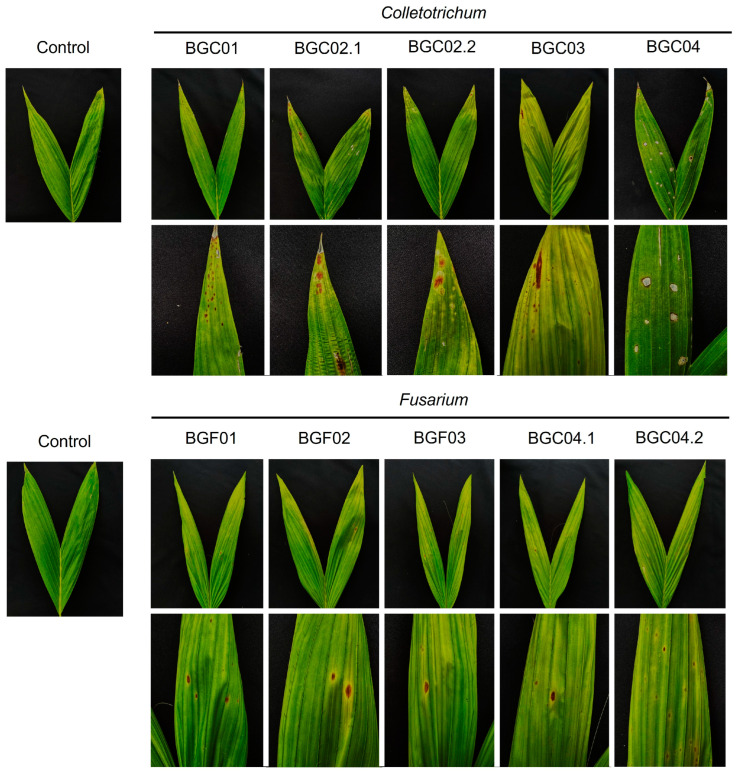
Symptoms of *Colletotrichum fructicola* (BGC02.2 and BGC03), *C. theobromicola* (BGC01, BGC02.1, and BGC04), and *Fusarium pernambucanum* (BGF01, BGF02, BGF03, BGF04.1, and BGF04.2) on healthy peach palm seedlings: Development of symptoms on healthy leaves (**above**) and zoomed-in view of symptoms (**below**).

**Figure 3 jof-11-00318-f003:**
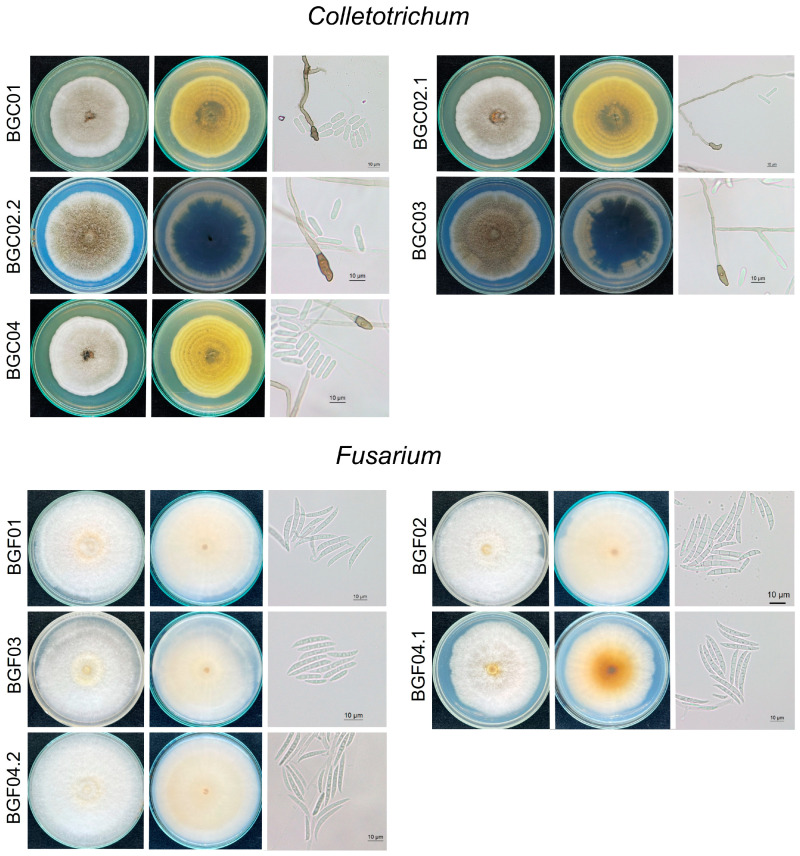
Morphological characteristics of *Colletotrichum fructicola* (isolates BGC02.2 and BGC03), *C. theobromicola* (isolates BGC01, BGC02.1, and BGC04), and *Fusarium pernambucanum*. Colony morphology on PDA: top view, bottom view, and microscopic features of conidia and appressoria (*Colletotrichum*) and for conidia (*Fusarium*) from left to right order in each plate.

**Figure 4 jof-11-00318-f004:**
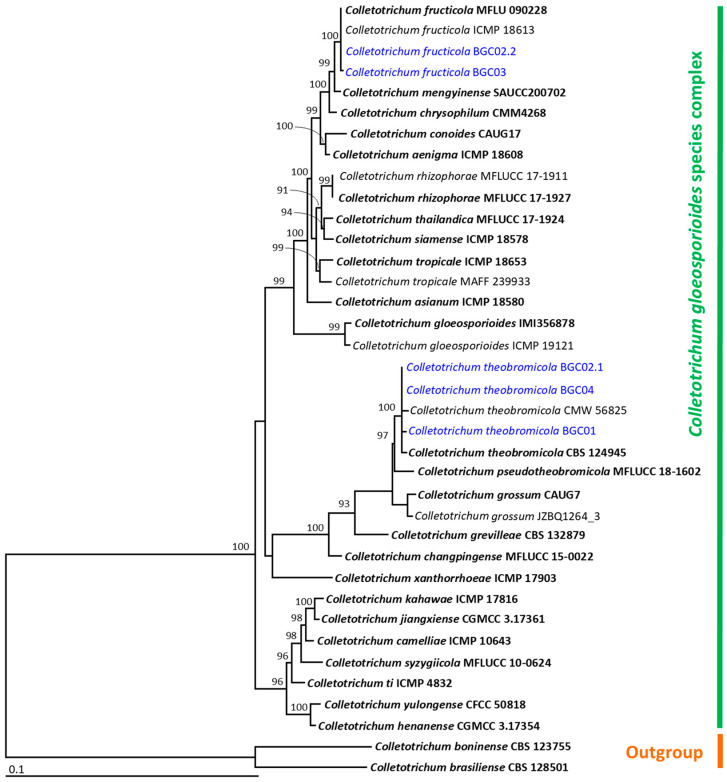
A phylogram derived from a maximum likelihood analysis of 37 fungal isolates of the combined ITS, *gapdh*, *cal*, *act*, and *tub* sequences. *Colletotrichum boninense* CBS 128501 and *C. brasiliense* CBS 123755 were set as the outgroup. The numbers above the branches represent bootstrap percentages, and values > 70% are shown. The scale bar represents the expected number of nucleotide substitutions per site. The fungal isolates obtained from this study are blue. Type species are bold.

**Figure 5 jof-11-00318-f005:**
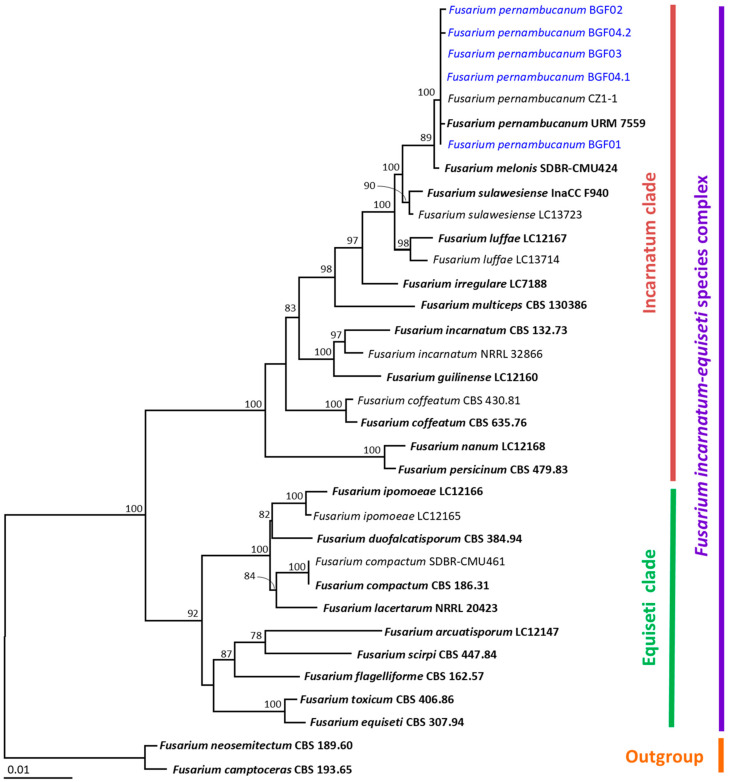
A phylogram derived from a maximum likelihood analysis of 30 fungal isolates of the combined *tef1-α*, *cal*, and *rpb2* sequences. *Fusarium camptoceras* CBS 193.65 and *F. neosemitectum* CBS 189.60 were set as the outgroup. The numbers above the branches represent bootstrap percentages, and values > 70% are shown. The scale bar represents the expected number of nucleotide substitutions per site. The fungal isolates obtained from this study are blue. Type species are bold.

**Figure 6 jof-11-00318-f006:**
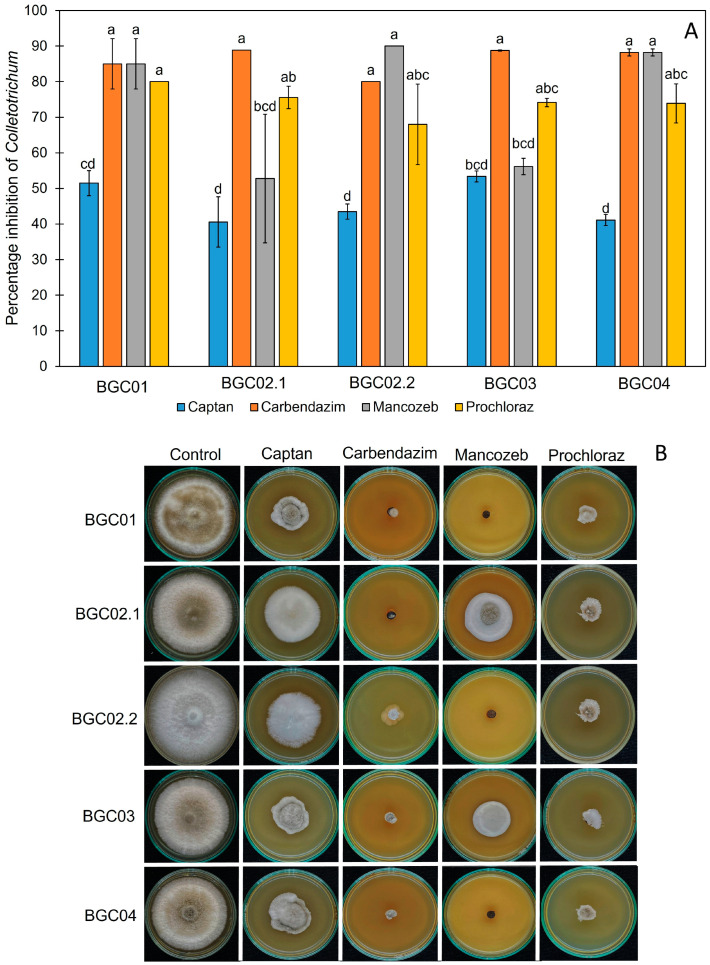
The effect of synthetic fungicides on the fungal growth of *Colletotrichum fructicola* (BGC02.2 and BGC03) and *C. theobromicola* (BGC01, BGC02.1, and BGC04). The percentage inhibitions of selected fungicides on the growth of *C. theobromicola* (**A**). Effects of fungicides on the growth of *C. theobromicola* on PDA of different fungicides (**B**). The values are given in the form of mean ± SD, and different letters indicate statistical differences among treatments according to Tukey’s Honestly Significant Difference test (*p* < 0.05). The concentration of each fungicide: captan (40 g/20 L), carbendazim (50 mL/20 L), mancozeb (50 g/20 L), and prochloraz (80 g/20 L).

**Figure 7 jof-11-00318-f007:**
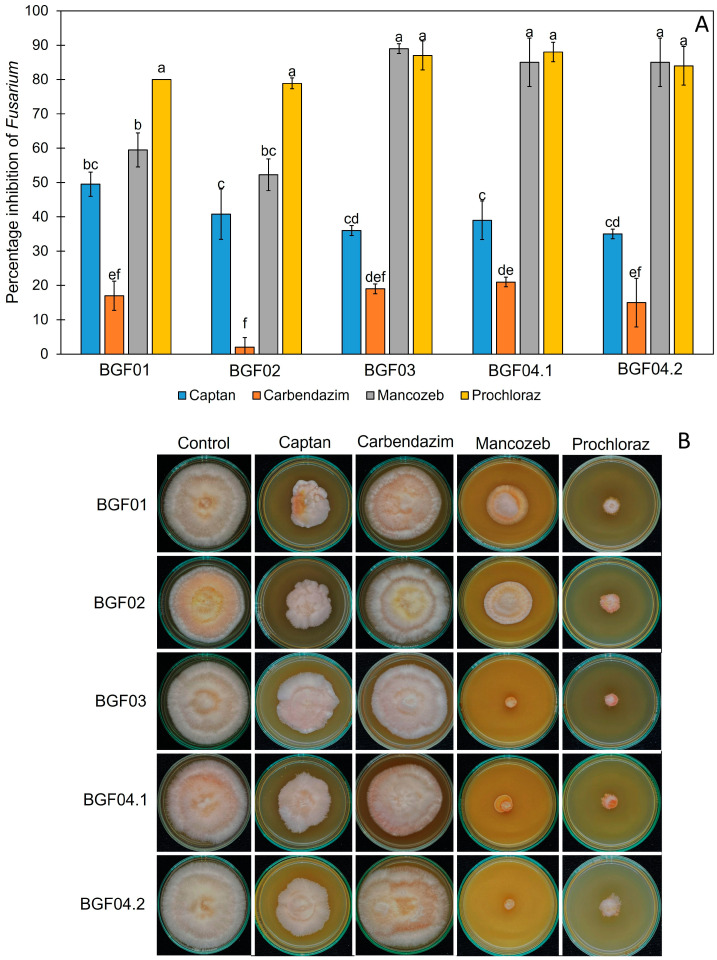
The effect of synthetic fungicides on the fungal growth of *Fusarium pernambucanum*. The *percentage* inhibitions of fungicides on the growth of *F. pernambucanum* (**A**). The effects of different fungicides on the growth of *F. pernambucanum* on on PDA (**B**). The values are given as mean ± SD, and different letters indicate statistical differences among treatments according to Tukey’s Honestly Significant Difference test (*p* < 0.05). The concentration of each fungicide: captan (40 g/20 L), carbendazim (50 mL/20 L), mancozeb (50 g/20 L), and prochloraz (80 g/20 L).

**Table 1 jof-11-00318-t001:** The GenBank accession numbers and the highest similarity values of sequences between the fungal strains in this study and type fungal species in the GenBank database.

Fungal Isolate	Gene	GenBank Accession Number	The Closely Related Type Fungal Species/Similarity Value (%)
BGC01	ITS	PQ764973	*Colletotrichum theobromicola* CBS 124945/100.00
	*gapdh*	PQ772077	*Colletotrichum theobromicola* CBS 124945/98.88
	*cal*	PQ772078	*Colletotrichum theobromicola* CBS 124945/99.59
	*act*	PQ772079	*Colletotrichum theobromicola* CBS 124945/97.43
	*tub*	PQ772080	*Colletotrichum theobromicola* CBS 124945/100.00
BGC02.1	ITS	PV298543	*Colletotrichum theobromicola* CBS 124945/100.00
	*gapdh*	PV334903	*Colletotrichum theobromicola* CBS 124945/98.87
	*cal*	PV334907	*Colletotrichum theobromicola* CBS 124945/99.59
	*act*	PV334911	*Colletotrichum theobromicola* CBS 124945/97.43
	*tub*	PV334915	*Colletotrichum theobromicola* CBS 124945/100.00
BGC02.2	ITS	PV290482	*Colletotrichum fructicola* MFU 090228/100.00
	*gapdh*	PV334904	*Colletotrichum fructicola* MFU 090228/100.00
	*cal*	PV334908	*Colletotrichum fructicola* MFU 090228/100.00
	*act*	PV334912	*Colletotrichum mengyinense* SAUCC 200702/100.00
	*tub*	PV334916	*Colletotrichum theobromicola* CBS 124945/100.00
BGC03	ITS	PV298544	*Colletotrichum fructicola* MFU 090228/100.00
	*gapdh*	PV334905	*Colletotrichum fructicola* MFU 090228/100.00
	*cal*	PV334909	*Colletotrichum fructicola* MFU 090228/100.00
	*act*	PV334913	*Colletotrichum mengyinense* SAUCC 200702/100.00
	*tub*	PV334917	*Colletotrichum fructicola* MFU 090228/100.00
BGC04	ITS	PV290486	*Colletotrichum theobromicola* CBS 124945/100
	*gapdh*	PV334906	*Colletotrichum theobromicola* CBS 124945/98.87
	*cal*	PV334910	*Colletotrichum theobromicola* CBS 124945/99.60
	*act*	PV334914	*Colletotrichum theobromicola* CBS 124945/97.47
	*tub*	PV334918	*Colletotrichum theobromicola* CBS 124945/100.00
BGF01	*cal*	PQ772082	*Fusarium melonis* SDBR-CMU424/100.00
	*tef1-α*	PQ772081	*Fusarium pernambucanum* URM 7559/99.56
	*rpb2*	PQ772083	*Fusarium pernambucanum* URM 7559/100.00
BGF02	*cal*	PV334919	*Fusarium melonis* SDBR-CMU424/100.00
	*tef1-α*	PV334923	*Fusarium pernambucanum* URM 7559/99.57
	*rpb2*	PV334927	*Fusarium pernambucanum* URM 7559/100.00
BGF03	*cal*	PV334920	*Fusarium melonis* SDBR-CMU424/100.00
	*tef1-α*	PV334924	*Fusarium pernambucanum* URM 7559/99.56
	*rpb2*	PV334928	*Fusarium pernambucanum* URM 7559/100.00
BGF04.1	*cal*	PV334921	*Fusarium melonis* SDBR-CMU424/100.00
	*tef1-α*	PV334925	*Fusarium pernambucanum* URM 7559/99.56
	*rpb2*	PV334929	*Fusarium pernambucanum* URM 7559/100.00
BGF04.2	*cal*	PV334922	*Fusarium melonis* SDBR-CMU424/100.00
	*tef1-α*	PV334926	*Fusarium pernambucanum* URM 7559/99.56
	*rpb2*	PV334930	*Fusarium pernambucanum* URM 7559/100.00

## Data Availability

The DNA sequence data obtained from this study have been deposited in GenBank under the following accession numbers: ITS (PQ764973, PV298543, PV290482, PV298544, and PV290486), *tef1-α* (PQ772081, PQ772081, PV334923, PV334924, PV334925, and PV334926), *cal* (PQ772078, PQ772082, PV334907, PV334908, PV334909, PV334910, PV334919, PV334920, PV334921, and PV334922), *act* (PQ772079, PV334911, PV334912, PV334913, and PV334914), *tub* (PQ772080, PV334915, PV334916, PV334917, and PV334918), *gadph* (PQ772077, PV334903, PV334904, PV334905, and PV334906), and *rpb2* (PQ772083, PV334927, PV334928, PV334929, and PV334930).

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
