# Peer review of "Fungal Pathogens of Peach Palm Leaf Spot in Thailand and Their Fungicide Sensitivity"

_jof, 2025, doi:10.3390/jof11040318_

Round 1
Reviewer 1 Report
1. the introduction at present provided infomation of other disease,few about leaf spot diseases on peach palm;
2. More experiment design and repeats needed. such as: A. Only A total of five symptomatic leaves were collected and Only one Colletotrichum sp. and one Fusarium sp. were assessed. For identify pathogen of a new disease. It has no enough repeates. B. For identify Colletotrichum sp. and Fusarium sp. as the pathogen of a new disease, innoculation with spore suspensions should be adopted.
1. 3.2. Pathogenicity Testing of Fungal Isolates. There will be several different leaf spot disease include anthracnose and others. The symptoms in Fig.2 and 1 are not the same.
2. key infomation for 3.5. are absent, such as concentrations
3. JUST AS indicated in Line 66-68, carbendazim is old fungicide with serious resistance. For,Colletotrichum sp. and Fusarium sp. , other newer fungicides such as DMIS should be tested.
Author Response
Major comments
- the introduction at present provided infomation of other disease,few about leaf spot diseases on peach palm;
Answer: According to an online database, leaf spot disease has been reported in oil palm (Elaeis guineensis), which belongs to the same family as peach palm (Bactris gasipaes). However, no specific reports on leaf spot diseases in peach palms were found. We have added this information into introduction part.
- More experiment designs and repeats are needed. such as: A. Only A total of five symptomatic leaves were collected and Only one Colletotrichum sp. and one Fusarium sp. were assessed. For identify pathogen of a new disease. It has no enough repeates. B. For identify Colletotrichum sp. and Fusarium sp. as the pathogen of a new disease, innoculation with spore suspensions should be adopted.
Answer: For A, we have collected more samples (20 leaf samples) and isolates for morphology studies (10 isolates) and molecular analyses of both Colletotrichum and Fusarium. Finally, we found at least 3 species causing leaf spots on peach palms. B, We have adopted to inoculate spore suspension on healthy leaves as indicated in Fig. 2
Detail comments
- 3.2. Pathogenicity Testing of Fungal Isolates. There will be several different leaf spot disease include anthracnose and others. The symptoms in Fig.2 and 1 are not the same.
Answer: The diseases found in natural infection were more severe than those observed in the laboratory (pathogenicity test), the natural infection may include leaf spot leaf blight, and anthracnose. This may be due to the high frequency of disease and physical factors involved in developing symptoms. After we adopted inoculation by spore suspension, the symptoms developed similarly to those observed in natural infection (Fig. 2).
- key infomation for 3.5. are absent, such as concentrations
Answer: We have added concentrations of each fungicide used in this study.
- JUST AS indicated in Line 66-68, carbendazim is old fungicide with serious resistance. For,Colletotrichum sp. and Fusarium sp. , other newer fungicides such as DMIS should be tested.
Answer: In this study, we would like to test common fungicides with different FRAC codes (indicate different modes of action) that are widely used in Thailand against leaf spot pathogens. Therefore, we selected captan (M04), carbendazim (1), and mancozeb (M03) and included the DMIS fungicide prochloraz (3) for testing against leaf spot pathogen as indicated in results Figs. 7 and 8.
Reviewer 2 Report
Make the suggested changes. Overall, your writing is of good to excellent quality, and contributes new knowledge to the field of knowledge on the range of hosts that fungi of these two genera attack. Congratulations.
jof-3415519 comments
Line 28: Italicize the words "In vitro"
Line 40: delete this phrase "in the Arecaceae family"
Line 97: Modify this text "(28±2°C)" like this, (28 ± 2°C)
Line 116: Modify this text “(n=20)” like this, (n = 20)
Line 141: Italicize the words “in vitro”
Line 189: Modify this text “4.53±1.29” like this, 4.53 ± 1.29
Line 194: Modify this text “(16.67±1.65)” like this, (16.67 ± 1.65)
Line 195: Modify this text “(4.34±0.47)” like this, (4.34 ± 0.47)
Line 198: Modify this text “7.35±2.21” like this, 7.35 ± 2.21
Line 204: Modify this text “(18.74±3.64)” like this, (18.74 ± 3.64)
Line 206: Modify this text “(7.98±2.87)” like this, (7.98 ± 2.87)
Line 217: Remove the space that exists between the numbers corresponding to the citations used.
Line 219: Remove duplicate word.
Line 227: Remove the % sign.
In works where the Bootstrap method of statistical support is used, neither the word nor the % sign is used. I recommend that authors eliminate it from their manuscript.
Line 294: Italicize the words “et al.”
Author Response
Make the suggested changes. Overall, your writing is of good to excellent quality, and contributes new knowledge to the field of knowledge on the range of hosts that fungi of these two genera attack. Congratulations.
Answer: Thank you for reviewing and giving valuable comments to improve this manuscript. We have carefully revised this manuscript with “RED TEXT”.
Line 28: Italicize the words "In vitro"
Answer: We have italicized “In vitro”.
Line 40: delete this phrase "in the Arecaceae family"
Answer: We have deleted this phrase.
Line 97: Modify this text "(28±2°C)" like this, (28 ± 2°C)
Answer: We have revised it as a suggestion.
Line 116: Modify this text “(n=20)” like this, (n = 20)
Answer: We have revised it as a suggestion.
Line 141: Italicize the words “in vitro”
Answer: We have italicized “In vitro”.
Line 189: Modify this text “4.53±1.29” like this, 4.53 ± 1.29
Answer: We have revised it as a suggestion.
Line 194: Modify this text “(16.67±1.65)” like this, (16.67 ± 1.65)
Answer: We have revised it as a suggestion.
Line 195: Modify this text “(4.34±0.47)” like this, (4.34 ± 0.47)
Answer: We have revised it as a suggestion.
Line 198: Modify this text “7.35±2.21” like this, 7.35 ± 2.21
Answer: We have revised it as a suggestion.
Line 204: Modify this text “(18.74±3.64)” like this, (18.74 ± 3.64)
Answer: We have revised it as a suggestion.
Line 206: Modify this text “(7.98±2.87)” like this, (7.98 ± 2.87)
Answer: We have revised it as a suggestion.
Line 217: Remove the space that exists between the numbers corresponding to the citations used.
Answer: We have revised it as a suggestion.
Line 219: Remove duplicate word.
Answer: We have revised it as a suggestion.
Line 227: Remove the % sign.
In works where the Bootstrap method of statistical support is used, neither the word nor the % sign is used. I recommend that authors eliminate it from their manuscript.
Answer: We have revised it as a suggestion.
Line 294: Italicize the words “et al.”
Answer: We have revised it as a suggestion.
Reviewer 3 Report
This manuscript is very well-written and concise, with a clear presentation of data. The images and phylogenetic trees are of high quality, and the identification appears to be reliable and thorough. The study provides valuable insights; however, I have some concerns that may impact its suitability as a full research paper for a journal of this impact factor.
The primary concern is the limited scope of the study, as it is based on only two isolates—one isolate of each specie from a single location. This narrow focus suggests that the manuscript might be better suited as a short communication rather than a full-length paper.
I would like the authors to address the following points for clarity and completeness:
1. Are these the first reports of these fungi in Thailand? If so, this should be clearly emphasized to highlight the novelty and significance of the findings.
2. How many types of mycelia were initially isolated? Were there other isolates besides these two that did not show pathogenicity?
3. In vitro fungicide tests, while useful, often do not reliably reflect in vivo effectiveness. In this case, the results are based on very limited data, with only one isolate, which is insufficient to draw conclusions that are both reliable and impactful. A more robust dataset would be necessary to establish findings with broader implications and make this title.
4. Did the pathogens produce distinct symptoms, and were there any differences in symptomatology? For instance, Fusarium and Colletotrichum are known to produce different types of discolorations or damage.
5. What is the reality of the associations described between these pathogens? The manuscript does not explain the nature of these associations or their significance. Have you tested their joint pathogenicity on the host or something else?
Addressing these questions and providing additional context on the reliability, novelty, and broader implications of the findings would help strengthen the study. While the manuscript presents solid and reliable data, the scope and depth of the work may not align with the expectations for a full research article in this journal. I leave this to the senior Editor to decide.
No additional comments
Author Response
This manuscript is very well-written and concise, with a clear presentation of data. The images and phylogenetic trees are of high quality, and the identification appears to be reliable and thorough. The study provides valuable insights; however, I have some concerns that may impact its suitability as a full research paper for a journal of this impact factor.
The primary concern is the limited scope of the study, as it is based on only two isolates—one isolate of each specie from a single location. This narrow focus suggests that the manuscript might be better suited as a short communication rather than a full-length paper.
Answer: Thank you for giving us valuable comments to improve this manuscript. We have increased number of collected samples, number of fungal isolated samples, and finally we found 3 fungal species causing leaf spot disease on peach palm.
I would like the authors to address the following points for clarity and completeness:
- Are these the first reports of these fungi in Thailand? If so, this should be clearly emphasized to highlight the novelty and significance of the findings.
Answer: Yes, this study represents the first report of Colletotrichum theobromicola, Colletotrichum fructicola, and Fusarium pernambucanum causing leaf spot disease on Bactris gasipaes in Thailand. To highlight the novelty and significance of our findings, we have clearly emphasized this aspect in the Introduction and Discussion sections. Additionally, we have provided relevant references to support the uniqueness of our study. Thank you for your valuable suggestion.
- How many types of mycelia were initially isolated? Were there other isolates besides these two that did not show pathogenicity?
Answer: There are two types of mycelia that were observed in this study; first mycelia that could not produce spores/conidia (this may be an endophyte) and second mycelia with spore/conidia. Based on the result of our study only the mycelia which could produce spores can cause leaf spots by pathogenicity test.
- In vitro fungicide tests, while useful, often do not reliably reflect in vivo effectiveness. In this case, the results are based on very limited data, with only one isolate, which is insufficient to draw conclusions that are both reliable and impactful. A more robust dataset would be necessary to establish findings with broader implications and make this title.
Answer: Thank you for your valuable feedback. We acknowledge that in vitro fungicide tests may not always accurately reflect in vivo effectiveness. However, they provide an essential preliminary screening for pathogen susceptibility. In this study, the fungicide sensitivity tests were conducted on a limited number of isolates due to resource constraints. We recognize that a more extensive dataset, including multiple isolates would be necessary to strengthen the generalizability of our findings. However, according to other reviewers’ comments, we have increased the number of collected samples, number of isolates, and number of species found in this manuscript. We tested the effect of synthetic fungicides in different modes of action (MoA) according to FRAC code, and finally found that the DMIS fungicide prochloraz showed positive results against three fungal species, which may guide farmers to select an appropriate fungicide to control leaf spot on peach palm.
- Did the pathogens produce distinct symptoms, and were there any differences in symptomatology? For instance, Fusarium and Colletotrichum are known to produce different types of discolorations or damage.
Answer: For the pathogenicity test, Colletotrichum produces symptoms at the tips of the leaf with circular concentric dark brown stained margin, whereas Fusarium showed circular to irregular brown to dark brown spots. However, as in natural infection, the symptoms seemed more severe than those of the pathogenicity test, and may combined leaf spot and anthracnose in the same leaf.
- What is the reality of the associations described between these pathogens? The manuscript does not explain the nature of these associations or their significance. Have you tested their joint pathogenicity on the host or something else?
Answer: Thank you for your insightful comment. In this study, we identified Colletotrichum theobromicola, Colletotrichum fructicola, and Fusarium pernambucanum as causal agents of peach palm leaf spot disease. However, we acknowledge that the manuscript did not explicitly discuss the nature of their associations. At present, our study primarily focuses on identifying individual pathogens and confirming their pathogenicity through separate inoculations. We did not conduct co-inoculation experiments to assess potential synergistic or antagonistic interactions among these fungi. However, the co-occurrence of multiple Colletotrichum species on the same host suggests a possible interaction, as reported in previous studies on Colletotrichum species complexes (Bhunjun et al., 2021). Due to the comment to no joint pathogenicity on the host in this manuscript, we have changed our title to “First Report of Multiple Fungal Pathogens Causing Leaf Spot Disease on Peach Palm (Bactris gasipaes) in Thailand and Their Fungicide Sensitivity”
Addressing these questions and providing additional context on the reliability, novelty, and broader implications of the findings would help strengthen the study. While the manuscript presents solid and reliable data, the scope and depth of the work may not align with the expectations for a full research article in this journal. I leave this to the senior Editor to decide.
Round 2
Reviewer 1 Report
The revised version has improved significantly.
- add species information for each isolate in Fig. 2-4 and Table 1;
- Provide concentrations for each fungicide in Fig. 7-8.
-
appressoria (c, g, k, o and s) should be presented with conidia in Fig.3.
Author Response
Major comments
The revised version has improved significantly.
Response: Thank you very much.
Detail comments
- add species information for each isolate in Fig. 2-4 and Table 1;
Response: We have added species information for each isolate in Figs. 2–3, fig. 2 was combined with fig. 3 and Table 1 were removed according to reviewer 3's suggestion.
- Provide concentrations for each fungicide in Fig. 7-8.
Response: We have added concentrations for each fungicide in Figs. 6–7 with yellow highlighted.
- appressoria (c, g, k, o and s) should be presented with conidia in Fig.3.
Response: We have added appressoria with conidia in Fig.3.
Reviewer 3 Report
The overall quality has improved; however, several details still require more careful attention. Abstract, some figure texts need to be rewritten for clarity, and certain sections of the manuscript remain wordy and would benefit from more concise phrasing.
These issues were not highlighted in the first round of review, as the initial focus was on addressing more significant concerns. The reviewer encourage you to carefully go through the entire manuscript again to ensure that the language is clear, the structure is logical, and the text is as concise as possible. If you are unsure how to structure or present specific parts of the manuscript—such as figures, or result interpretations— strongly recommend consulting recent publications in reputable journals as a guide.
Thank you for your continued efforts.
Comments are in the attached file

Author Response
Major comments
The overall quality has improved; however, several details still require more careful attention. Abstract, some figure texts need to be rewritten for clarity, and certain sections of the manuscript remain wordy and would benefit from more concise phrasing.
These issues were not highlighted in the first round of review, as the initial focus was on addressing more significant concerns. The reviewer encourage you to carefully go through the entire manuscript again to ensure that the language is clear, the structure is logical, and the text is as concise as possible. If you are unsure how to structure or present specific parts of the manuscript—such as figures, or result interpretations— strongly recommend consulting recent publications in reputable journals as a guide.
Thank you for your continued efforts.
Response: We sincerely appreciate the valuable feedback and constructive comments provided by the reviewers. We have carefully addressed all suggestions and revised the manuscript accordingly. Below, we provide a point-by-point response to each comment, highlighting the changes made in the revised manuscript (blue highlighted).
For the abstract: We have rewritten the abstract to enhance clarity and ensure conciseness while maintaining all essential information. The revised abstract now succinctly presents the key findings and significance of the study
For the description of disease symptoms: We have revised the description of disease symptoms to be more concise and precise. The updated version removes redundant phrases and improves readability while retaining all necessary details.
For figure captions: The figure captions have been revised for clarity and consistency. We have also ensured that figure labeling is clearer, following best practices observed in high-impact journals.
For first report of these fungal species: We have explicitly stated in the discussion section that this study represents the first report of Colletotrichum theobromicola and Fusarium pernambucanum causing leaf spot disease on peach palm, both globally and in Thailand. We have also added relevant citations to provide context regarding their known distributions.
For fungicide sensitivity assay: We have clarified in the discussion section that the fungicide sensitivity assay conducted in this study was a preliminary in vitro screening. We also emphasized the need for future field-based validation to confirm its practical applications.
For some sections of the manuscript remain wordy and require more concise phrasing: We have thoroughly reviewed the manuscript and revised sections that were overly wordy. Unnecessary repetition has been removed, and the phrasing has been refined for conciseness without compromising clarity.
Detail comments
Comments are in the attached file
Response: We have revised all suggestions with blue highlighting text.
We greatly appreciate the reviewers' insightful comments, which have helped improve the quality of our manuscript. We hope the revised version meets the expectations of the reviewers and the editorial team. Please let us know if further modifications are required.
Round 3
Reviewer 3 Report
All of the suggested comments have been addressed and accepted by the authors.
I have no further comments
I appreciate the authors’ efforts to address the previous comments, and I acknowledge
that improvements have been made.